# Tri-Criterion Model for Constructing Low-Carbon Mutual Fund Portfolios: A Preference-Based Multi-Objective Genetic Algorithm Approach

**DOI:** 10.3390/ijerph17176324

**Published:** 2020-08-31

**Authors:** Adolfo Hilario-Caballero, Ana Garcia-Bernabeu, Jose Vicente Salcedo, Marisa Vercher

**Affiliations:** 1Institute of Control Systems and Industrial Computing (ai2), Universitat Politècnica de València, 46022 Valencia, Spain; ahilario@upv.es (A.H.-C.); jsalcedo@upv.es (J.V.S.); 2Campus of Alcoi, Universitat Politècnica de València, 03801 Alcoi, Spain; maverfer@upv.es

**Keywords:** genetic algorithms, low-carbon economy, multi-objective optimization, sustainable finance, investor’s preferences

## Abstract

Sustainable finance, which integrates environmental, social and governance criteria on financial decisions rests on the fact that money should be used for good purposes. Thus, the financial sector is also expected to play a more important role to decarbonise the global economy. To align financial flows with a pathway towards a low-carbon economy, investors should be able to integrate into their financial decisions additional criteria beyond return and risk to manage climate risk. We propose a tri-criterion portfolio selection model to extend the classical Markowitz’s mean-variance approach to include investor’s preferences on the portfolio carbon risk exposure as an additional criterion. To approximate the 3D Pareto front we apply an efficient multi-objective genetic algorithm called ev-MOGA which is based on the concept of ε-dominance. Furthermore, we introduce a-posteriori approach to incorporate the investor’s preferences into the solution process regarding their climate-change related preferences measured by the carbon risk exposure and their loss-adverse attitude. We test the performance of the proposed algorithm in a cross-section of European socially responsible investments open-end funds to assess the extent to which climate-related risk could be embedded in the portfolio according to the investor’s preferences.

## 1. Introduction

Climate change will pose a challenge for the financial sector seeking a balance between purely financial goals—looking for high returns—and sustainability making a positive impact on the environment and society. Since 2015, by adopting the Paris Agreement on climate-change and the UN 2030 for Sustainable Development, there has been a clear commitment, especially in the European Union, to align financial flows with a pathway towards a low-carbon, more resource-efficient and sustainable economy. In 2018, the EU launched and Action Plan to set out a strategy for sustainable finance, that is, “the process of taking due account of environmental, social and governance (ESG) considerations in investment decision-making, leading to increased investments in longer-term and sustainable activities” [1]. As stated in this report, to date, environmental and climate risks had not been appropriately considered by the financial sector, which is why if the EU wants to reorient capital flows to a more sustainable economy, environmental and social goals will have to be included in the financial decision-making. To this end, the Markets in Financial Instruments Directive (MIFID II) and the Insurance Distribution Directive (IDD) provides that investment firms and insurance distributors should ask their clients’ investments objectives as regard sustainability and take their preferences into account when providing financial advice. This implies that investors should be able to integrate into their financial decisions additional criteria beyond return and risk and then to extend the classical bi-criterion portfolio selection problem based on Markowitz’s mean-variance approach [2] by adding one more criterion.

In the literature, tri-criterion portfolio selection problems have been addressed by several authors making use of multicriteria decision-making problems (MCDM). While, the first attempts to deal with three-criterion portfolio selection problems were made by means of exact methods, several heuristic approaches such as multi-objective evolutionary algorithms (MOEAs) have been employed to solve multi-objective portfolio optimization problems. A better understanding of recent MCDM approaches to deal with tri-criterion portfolio selection problems by reviewing the literature on exact methods and multi-objective genetic algorithms techniques is undertaken in Section 2. From this review, we find there is a clear gap in the literature in the use of MOEAs to asses and manage the impact of carbon risk related preferences in the portfolio selection problem.

In line with the goals of the Paris Agreement, the financial flows should be consistent with a pathway towards low greenhouse gas emissions. In recent years, in response to an increasing climate-conscious financial products demand, Morningstar, the most important information provider in the mutual fund industry, introduced the low carbon designation (LCD) eco-label [3]. This new mutual fund eco-label helps investors to easily recognize which mutual funds are aligned with the transition to a low-carbon economy [4]. The LCD is composed of two indices, the Carbon Risk score and the Fossil Fuel involvement. In our research, we only consider the fund-level Carbon Risk score, which is obtained by weighting the firm-level exposure and management of material carbon issues. As Krueger et al. [5] highlighted, institutional investors increasingly address climate-change related risks and they are also viewed as catalysing driving firms to meet the reduced emission target. Thus, the mutual fund industry, and in particular institutional investors is an ideal setting to test our proposal.

The main contributions of this paper are summarized as follows: (i) We propose an approach based on a recent multi-objective genetic algorithm called ev-MOGA [6] to assess and manage the impact of climate-change related risk including as a third objective a new measure of carbon risk based on the LCD benchmark provided by Morningstar. The proposed ev-MOGA ensure the convergence of uniformly distributed solutions with no conditions related to the type of the objective function. (ii) Once we obtain the approximated 3D non-dominated surface for the tri-criterion multi-objective problem, we introduce a-posteriori preference handling approach including the investor’s preferences about carbon risk and risk aversion to come up with a single solution. Thus, according to the investor’s preferences, we provide a bounded ε-Pareto front with information about minimum carbon risk, minimum risk aversion, and maximum return. With this tool, the investors could address more effectively with tri-criterion problems in the sense that they can easily visualize and find a desirable trade-off among conflicting objectives.

The rest of the paper is structured as follows. A review of the literature about the tri-criterion portfolio selection problem addressed either by exact or heuristic methodologies is presented in Section 2. In Section 3, the proposed tri-criterion genetic multi-objective evolutionary algorithm for constructing low carbon portfolios is formulated. In addition, we provide a-posteriori approach to integrate carbon risk related preferences in financial decision-making. In Section 4, we present the numerical results obtained by the application of the ev-MOGA using different investor profiles for a data set of European socially responsible investments (SRI) open-end funds. The discussion of the results is presented in Section 5. Finally, Section 6 concludes the paper.

## 2. Literature Review

The idea of determining the Pareto efficient frontier in portfolio selection from a mean-variance (M-V) optimization was initially conceived by Markowitz [2]. The essence of the M-V model is that risk is the investor’s main concern and he/she tries to minimize risk for the desired level of expected returns. Over the years, the Markowitz’s model has been extended either through more complex risk measures or through additional constraints, and in recent years through the possibility to include additional objectives. In this context, two main approaches to deal with the extended portfolio optimization problem can be found: (i) Exact methods or (ii) Heuristic methodologies. In what follows, we provide an integrated review of the theoretical and practical works that have used exact methods or heuristic approaches for extended M-V portfolio selection.

### 2.1. M-V Extended Approaches by Exact Methods

Since the early 1970s, several authors have attempted to expand the classical bi-criteria portfolio selection model beyond the expected return and variance with exact methods. Three main groups of studies can be identified by dealing with this problem. The first group of authors have expanded the Markowitz’s model by introducing additional constraints such as cardinality, round lots or buy-in threshold [7,8,9,10]. Alternative risk measures such as down-side risk measures or CVaR have been proposed in the second group of studies [11,12,13]. A literature review on risk measures in terms of computational comparison is conducted in Mansini et al. [14].

Not until the 20th century was the idea of additional objectives was further boosted by the third group of studies. A tri-criterion non dominated surface can be found in [15,16,17] using a constrained linear program (QCLP) approach by solving a quad-lin-lin optimization problem where the third objective is linear. By defining several measures of liquidity in Lo et al. [18] a three-dimensional mean-variance-liquidity frontier is constructed. A general framework for computing the non-dominated surface in tri-criterion portfolio selection that extends the Markowitz’s portfolio selection approach to an additional linear criterion (dividends, liquidity or sustainability) is addressed in Hirschberger et al. [15]. By solving a quad-lin-lin program, they provide an exact method for computing the non-dominated surface that can outperform standard portfolio strategies for multicriteria decision makers. An empirical application where the third criterion is sustainability is developed to illustrate how to compose the non-dominated surface.

In Utz et al. [16] sustainability is included as the third criterion to obtain the variance-expected return-sustainability efficient frontier in order to explain how the sustainable mutual fund industry can increase its levels of sustainability. The tri-criterion non-dominated surface is computed through the Quadratic Constrained Linear Program (QCLP) approach, and from the experimental results, it can be concluded that there was room to expand the sustainability levels without hampering the levels of risk and return.

However, the existing proposals based on exact procedures to solve tri-criteria portfolio selection problems have limited capabilities when the third objective is non-linear. In such cases, heuristic techniques have been recently applied to solve multi-objective problems and to provide fair approximations of the Pareto front.

### 2.2. Moeas and the Extended M-V Portfolio Optimization Problem

The increasing complexity of financial decision-making problems has led researchers to apply heuristic procedures inspired by biological processes such as multi-objective evolutionary algorithms (MOEAs). Suggested at the beginning of the 1990s, MOEAs have been applied in several fields including finance, and in particular to solve the portfolio selection problem [19,20]. These techniques provide satisfactory approximations of the efficient frontier even when the problem involves non-convexity, discontinuity or non-integer variables. In Arnone et al. [21], a MOEA was proposed for the first time for optimal portfolio selection by using lower partial moments as a measure of risk. The first attempts to propose MOEAs as an extension of the M-V model aimed at considering additional constraints such as, cardinality, lower and upper bounds, transaction costs, transaction round lots, non-negativity constraints or sector capitalization constraints [10,22,23,24,25,26,27,28,29,30]. A review of the state of the art of MOEAs in portfolio selection can be found in Metaxiotis and Liagkouras [31]. In Silva et al. [32] a unified method with the ability of tackling several constraints (floor and ceiling, cardinality, round-lot and pre-assignment) is presented.

Another group of researchers have also tried to propose alternative risk measures to variance, the most popular being: semi-variance, value at risk (VaR) and conditional value at risk (CVaR), the lower partial moments (LPM), the Expected Shortfall, the Skewness, and Risk parity [33,34,35,36,37]. Other approaches used MOEAs with copula functions to consider the dependence structure of asset returns, see for example Babaei et al. [38]. Some scholars have been focused on preference-based algorithms for portfolio selection to direct the search of the non-dominated solutions towards de region of interest [39].

Regarding the number of objectives, while the two-objective case is the most widely used among the authors, the tri-objective problem has risen in popularity in the last few years. A tri-objective optimization problem is proposed in Anagnostopoulos and Mamanis [40] to find the trade-off between risk, return and the number of securities in the portfolio. In this paper, the authors compare three evolutionary multi-objective optimization techniques for finding the best trade-off between risk, return and the cardinality of the portfolio. A multi-swarm multi-objective optimizer based on p-optimality criteria called p-MSMOEAs is proposed with three-objectives in Hu et al. [41]. In Rangel-González et al. [42], a fuzzy multi-objective particle swarm optimization (FOMOPSO) algorithm that implements a fuzzy controller is implemented to solve a three-objective portfolio optimization problem.

A recent approach, based on the elitist evolutionary multi-objective optimization algorithm called ev-MOGA [6], is adopted in Garcia-Bernabeu et al. [43] to derive the non-dominated mean-variance-sustainability surface. While in the previous study the third objective was an overall score of sustainability including an integrated measure of ESG issues, we now focus on climate-change related objectives such as to reduce low carbon emissions. To the best of our knowledge, this is the first time that a MOEA has been used to assess and manage the impact of climate-related risks in the portfolio selection problem as a response to the upcoming more stringent climate policies.

## 3. The Tri-Criterion Multi-Objective Approach by Ev-Moga to Manage Carbon Risk Exposure

During the last two decades, MOEAs for portfolio management have attracted scholars and practitioners attention as stated in Section 2.2. Next, some previous notions on multi-objective optimization and genetic multi-objective optimization techniques are provided.

### 3.1. Background on Multi-Objective Optimization and Ev-Moga

Multi-objective optimization is an important subclass of multiple criteria decision making techniques involving more than one objective function to be optimized simultaneously. Since the conflict degree between the objectives makes it impossible to find a feasible solution that simultaneously optimizes all the objective functions, there is a set of Pareto optimal solutions denoted as Pareto front at which none of the objectives can be improved without deteriorating at least one of the others. In general, a multi-objective optimization problem is stated as follows:(1)$$\begin{array}{cc} \; & \underset{w}{\mathrm{minimize}}\;f\left(w\right)={\left[\;f_{1}\left(w\right),f_{2}\left(w\right),\;\ldots ,\;f_{m}\left(w\right)\;\right]}^{T},\\ \; & \mathrm{subject}\;\mathrm{to}\;w\in S, \end{array}$$
where the vector $w={\left[\omega_{1},\omega_{2},\;\ldots ,\;\omega_{n}\right]}^{T}$ is a *n*–parameter set included in the decision space *S*, and $f_{i}\left(w\right):R^{n}\to R$, i=1,…,m, are the objectives to be minimized at the same time.

In recent years MOEAs have been widely accepted as useful tools for solving real-world multi-objective problems. Within MOEAs several powerful stochastic search techniques that mimic Darwinian principles of natural selection are included. In this study, we focus on the ev-MOGA algorithm proposed in Herrero [6], which combines the concept of Pareto optimality and ε-dominance due to Laumanns et al. [44], thus providing an approximated ε-Pareto set.

**Definition** **1.**
*Dominance: Let $w^{1},\;w^{2}\in R^{n}$ be two feasible solutions, and let $f\left(w^{1}\right),\;f\left(w^{2}\right)\in R^{m}$ be their image solutions in the objective space. Then, assuming that the objective functions have to be minimized, $w^{1}$ is said to dominate $w^{2}$, denoted as $f\left(w^{1}\right)≺f\left(w^{2}\right)$, iff:*
(2)$$\begin{array}{cc} \; & \forall i\in \left\{1,\;\ldots ,\;m\right\}\;:\;f_{i}\left(w^{1}\right)\leq f_{i}\left(w^{2}\right)\\ \; & \exists \;j\in \left\{1,\;\ldots ,\;m\right\}\;:\;f_{j}\left(w^{1}\right)<f_{j}\left(w^{2}\right) \end{array}$$


**Definition** **2.**
*Pareto front: Let $\Omega \subseteq R^{n}$ be a set of vectors of feasible solutions with f(Ω) as their image solutions. Then the Pareto front $f(\Omega_{P})$ is defined as follows: $f(\Omega_{P})$ contains all vectors $f\left(w^{u}\right)\in f\left(\Omega \right)$ that are not dominated by any vector $f\left(w^{v}\right)\in f\left(\Omega \right)$, i.e.,*
(3)$$f\left(\Omega_{P}\right):=\left\{\;f\left(w^{u}\right)\in f\left(\Omega \right)\;\;\;|\;\;\;∄\;f\left(w^{v}\right)\;:\;f\left(w^{v}\right)≺f\left(w^{u}\right)\;\right\}$$
*From this definition, it is easily to deduce that any vector $f\left(w^{v}\right)\in f\left(\Omega \right)\;\setminus \;f\left(\Omega_{P}\right)$ is dominated by at least one vector $f\left(w^{u}\right)\in f\left(\Omega_{P}\right)$, i.e.,*
$$\forall \;f\left(w^{v}\right)\in f\left(\Omega \right)\;\setminus \;f\left(\Omega_{P}\right):\exists \;f\left(w^{u}\right)\in f\left(\Omega_{P}\right)\;\;|\;\;f\left(w^{u}\right)≺f\left(w^{v}\right)$$


**Definition** **3.**
*ε-dominance: Let $w^{1},\;w^{2}\in R^{n}$ be two feasible solutions, an let $f\left(w^{1}\right),\;f\left(w^{2}\right)\in R_{+}^{m}$ be their image solutions in the objective space. Then, assuming that the objective functions have to be minimized, $w^{1}$ is said to ε-dominate $w^{2}$ for some ε>0, denoted as $f\left(w^{1}\right){≺}_{\varepsilon }f\left(w^{2}\right)$, iff:*
(4)$$\forall \;i\in \left\{1,\;\ldots ,\;m\right\}\;:\;\left(1-\varepsilon \right)\;\cdot \;f_{i}\left(w^{1}\right)\leq f_{i}\left(w^{2}\right)$$


**Definition** **4.**
*ε-approximate Pareto front: Let $\Omega \subseteq R^{n}$ be a set of feasible solution vectors with f(Ω) as their image solutions. Then, $f({\hat{\Omega }}_{\;P}^{*})$ is called a ε-approximate Pareto front if any vector $f\left(w^{v}\right)\in f\left(\Omega \right)\;\setminus \;f\left({\hat{\Omega }}_{\;P}^{*}\right)$ is ε-dominated by at least one vector $f\left(w^{u}\right)\in f\left({\hat{\Omega }}_{\;P}^{*}\right)$, i.e.,*
(5)$$\forall \;f\left(w^{v}\right)\in f\left(\Omega \right)\;\setminus \;f\left({\hat{\Omega }}_{\;P}^{*}\right):\exists \;f\left(w^{u}\right)\in f\left({\hat{\Omega }}_{\;P}^{*}\right)\;\;|\;\;f\left(w^{u}\right){≺}_{\varepsilon }f\left(w^{v}\right)$$
*The set of all ε-approximate Pareto fronts of f(Ω) is denoted as the ε-Pareto front $f({\hat{\Omega }}_{\;P})$.*


The most outstanding feature of this algorithm is that the optimal solutions are distributed uniformly across the ε-Pareto front. To this end, the ε-Pareto front is split into a fixed number of boxes forming a grid, so that the algorithm ensures that just one solution is stored by one box. The size of the boxes is determined by the value of $\varepsilon_{i}$, which is calculated as follows:(6)$$\varepsilon_{i}=\frac{{f_{i}}^{*}-{f_{i}}_{*}}{n_{\mathrm{box}}}$$
where, $f_{i}^{*}$ and $f_{i*}$ correspond to the maximum and minimum value of the objective function $f_{i}$, and $n_{\mathrm{box}}$ is the number of boxes. In addition, ev-MOGA is able to adjust the width of $\varepsilon_{i}$ dynamically and prevent solutions belonging to the extremes of the front from being lost.

For solving the ev-MOGA, the main population $P_{k}$ whose size is $N_{\mathrm{indP}}$ explores the searching space *S* defined by the multi-objective problem during a number *k* of iterations. In the archive population $A_{k}$ the $\varepsilon_{i}$–nondominated solutions are stored. Then, at the end of the iteration process, $A_{k}$ is an ε-approximate Pareto front $f({\hat{\Omega }}_{\;P}^{*})$, with ${\left(n_{\mathrm{box}}\right)}^{m-1}$ as the maximum number of solutions, where *m* is the number of objectives. Furthermore, in the case that more than one ε-nondominated solution is detected, the solution that prevails in $A_{k}$ will be the one that is closest to the center of the box. Next, the new individuals obtained by crossover or mutation with probability of crossing/mutation $P_{\;}c\;/\;m$ are included in the auxiliary population $G\;A_{k}$. Before running the algorithm, the following parameters should be defined by the analyst:$N_{\mathrm{indP}}=$ Size of the main population.$N_{\mathrm{indGA}}=$ Size of the auxiliary population.$k_{\max}=$ Maximum algorithm iterations.$P_{\;}c\;/\;m=$ Probability of crossing/mutation.$n_{\mathrm{box}}=$ Number of boxes.

The main advantage of ev-MOGA is that they generate good approximations of a well-distributed Pareto front in a single run and within limited computational time. The original ev-MOGA algorithm is available at Matlab Central [45]: ev-MOGA in Matlab Central.

### 3.2. The Ev-Moga Tri-Criterion Portfolio Selection

In this study, beyond risk and return, we wish to consider an additional objective that minimizes the carbon risk exposure of a portfolio. Then, by introducing a third objective into the portfolio optimization model the efficient frontier becomes a surface in the three-dimensional space. The tri-criterion portfolio selection problem where the objectives are the risk of the portfolio, the returns, and the portfolio carbon risk exposure can be mathematically formulated as follows: (7)$$\begin{array}{cc} \min f_{1}\left(w\right)= & \sum_{i=1}^{N}\sum_{j=1}^{N}\omega_{i}\;\omega_{j}\;\sigma_{ij} \end{array}$$(8)$$\begin{array}{cc} \max f_{2}\left(w\right)= & \sum_{i=1}^{N}\omega_{i}\;\mu_{i} \end{array}$$(9)$$\begin{array}{cc} \min f_{3}\left(w\right)= & \sum_{i=1}^{N}\omega_{i}\;c_{i} \end{array}$$(10)$$\begin{array}{cc} \mathrm{subject}\;\mathrm{to}\; & \sum_{i=1}^{N}\omega_{i}=1,\;0\leq \omega_{i}\leq \omega_{i,\max} \end{array}$$
where *N* denotes the available assets, $\mu_{i}$ is the expected return of asset *i* (i=1,2,…,N), $\sigma_{ij}$ is the covariance between asset *i* and *j*. In addition, $c_{i}$ is the carbon risk score and $\omega_{i}$ denotes the proportion of asset *i* in the portfolio. Algorithm 1 details the ev-MOGA steps to obtain the well-distributed ε-approximate Pareto front.

**Algorithm 1** Tri-criterion ev-MOGA algorithm based on [6]1:Set k=0.2:Initialize the population of candidate solutions $P_{0}$ and set $A_{0}=\emptyset$3:Conduct the multi-objective evaluation of portfolios from $P_{0}$ using Equations (Equation 7)–(Equation 10)4:Detect the ε-nondominated portfolios from $P_{0}$ and store in the archive $A_{0}$5:
**while**
$k\leq k_{\max}$
**do**
6:  Generate the auxiliary population $G\;A_{k}$ from the main population $P_{k}$ and the     archive population $A_{k}$ following this procedure:7:  **for**
$j\leftarrow 1,N_{\mathrm{indGA}}/2$
**do**8:    Randomly select two portfolios $X^{P}$ and $X^{A}$ from $P_{k}$ and $A_{k}$, respectively9:    Generate a random number u∈[0,1]10:    If $u>P_{\;}c\;/\;m$, $X^{P}$ and $X^{A}$ are crossed over by means of the extended linear recombination      technique, generating two new portfolios for $G\;A_{k}$11:    If $u\leq P_{\;}c\;/\;m$, $X^{P}$ and $X^{A}$ are mutated using random mutation with Gaussian distribution       and then included in $G\;A_{k}$12:  **end for**13:  Evaluate population $G\;A_{k}$ using the tri-criterion multi-objective portfolio model defined by Equations (Equation 7)–(Equation 10).14:  Check which portfolios in $G\;A_{k}$ must be included in $A_{k+1}$ on the basis of their location     in the objective space. $A_{k+1}$ will contain all the portfolios from $A_{k}$ that are     not ε-dominated by elements of $G\;A_{k}$, and all the portfolios from $G\;A_{k}$ which are     not ε-dominated by elements of $A_{k}$15:  Update population $P_{k+1}$ with portfolios from $G\;A_{k}$. Every portfolio $X^{G\;A}$ from $G\;A_{k}$ is compared     with a portfolio $X^{P}$ that is randomly selected from the portfolios in $P_{k}$. $X^{G\;A}$ will replace     $X^{P}$ in $P_{k+1}$ if it dominates $X^{P}$. Otherwise, $X^{P}$ will not be replaced16:  k←k+117:
**end while**


### 3.3. Defining A-Posteriori Preferences for Each Investor Profile

With the previous multi-objective optimization design a vast region of the tri-objective whole Pareto front is generated. Even though it is true that the non-dominated surface allows us to better understand the trade-off between the three objectives, this solution does not provide a useful tool from the user’s perspective. To come up with a single solution, we assume that the decision-maker is available to take part in the solution process. According to [46,47] the articulation of preferences may be done either before (a-priori), during (progressive), or after (a-posteriori) the optimization process. In what follows, we assume that once the investor has seen an overview of the Pareto optimal solutions, they takes part in the final solution. Thus, we propose an a-posteriori approach.

The analyst supporting a-posteriori methodology has to inform the decision-maker either providing a list of solutions or to provide a visualization of the Pareto front [48]. In a tri-objective case, two main approaches have been used to visualize the Pareto frontier: (i) three-dimensional graph, and (ii) decision maps. However, a new graphical visualization called Level Diagram is proposed in Blasco et al. [49] to represent n-dimensional Pareto fronts. The Level Diagrams tool also allows the incorporation of decision-maker preferences, and it offers an excellent tool to help in the decision-making process.

In our proposal, information on carbon risk and risk aversion related preferences is given by the investor, which determine the upper bounds for the ε-Pareto front. Let us denote the vector for the preferences about green investments defined by the carbon risk score objective function in Equation (Equation 9) as $P_{g}$, and the preferences for the loss aversion attitude defined by Equation (Equation 7) as $P_{r}$.

Concerning the carbon risk related preferences, we consider three types of green investor profile. They are defined as follows:Strong green investor. This profile is defined by low level of allowed carbon risk $p_{g}^{s}$.Moderate green investor. This profile is defined by medium level of allowed carbon risk $p_{g}^{m}$.Weak green investor. This profile is defined by high level of allowed carbon risk $p_{g}^{w}$.

Thus, the reference vector for the green investor could be stated as follows:(11)$$P_{g}=\left[p_{g}^{s},\;p_{g}^{m},\;p_{g}^{w}\right]$$

Concerning the investor’s loss aversion attitude, we consider three types of risk aversion investor profile. They are defined as follows:Conservative investor. This profile is characterized by investing in lower-risk securities, namely, a high loss aversion attitude $p_{r}^{c}$.Cautious investor. This profile is defined by medium risk tolerance, and consequently a moderate loss aversion attitude $p_{r}^{k}$.Aggressive investor. It includes investors that actively seek stocks with higher risk, but a chance for higher reward, that is a low loss aversion attitude $p_{r}^{a}$.

Thus, the reference vector regarding the risk aversion could be stated as follows:(12)$$P_{r}=\left[p_{r}^{c},\;p_{r}^{k},\;p_{r}^{a}\right]$$

Figure 1a shows a three-dimensional representation of the ε-approximate Pareto front generated by ev-MOGA for the tri-criterion problem defined in Equations (Equation 7)–(Equation 10). The point’s color is faded from gray to blue: the more blue the color is, the more close to the global optimal point we are, which is colored in red. The optimal point is the one with the minimum normalized distance to the ideal. As may be seen, ev-MOGA provides uniformly distributed efficient portfolios along the ε-Pareto front.

The impact of considering the investor’s preferences (Equation 11) and (Equation 12) into the decision-making process is examined in Figure 1b. The new colored area, faded from blue to green, defines the investor’s region of interest throughout a bounded ε-Pareto front according to the previously defined preferences. Furthermore, in red we have colored the portfolios that can be proposed to the investor to maximize the returns (max ret), minimize the risk (min risk) or minimize the emission risk (min carb). The red point located in the green area is associated with the optimal portfolio that simultaneously achieves the three objectives within the investor’s region of interest.

## 4. Empirical Application

We use a set of monthly returns on 22 institutional SRI European open-end funds offered in Spain for the period 2009–2019. The empirical information includes the time series of 120 monthly returns and the carbon risk indices. As a previous step the expected return vector $\mu ={\left(\mu_{1},\;\ldots ,\;\mu_{22}\right)}^{T}$ and the covariance matrix $\Sigma =[\sigma_{ij}]$, i,j=1,…,22 are computed. For the carbon risk score $c_{i}$, we use the Morningstar^®^ Portfolio Carbon Risk Score, which indicates the risk that companies face from the transition to a low-carbon economy. In this set, scores $c_{i}$ range from 0 to 10, where lower scores are better, indicating lower carbon risk levels. All the numerical information to be used on this opportunity set comes from Morningstar database.

Table 1 shows the parameter setting applied to the ev-MOGA algorithm. The size of the main population is $N_{\mathrm{indP}}=10^{4}$, while the population of the archive $A_{k}$ is $N_{\mathrm{indGA}}=500$. For the probability of crossing/mutation we select $P_{\;}c\;/\;m=0.2$. Finally, the space of each objective function has been divided in 300 boxes.

The proposed approach was coded in Matlab^®^ R2020a, and it was executed on an Intel i7 3.20 GHz, 20 GB (2667 MHz DDR4) of RAM. As a result, the ev-MOGA algorithm provided an ε-approximate Pareto front of $N_{p}=7361$ solutions stored in two data arrays: first, an $\left(N_{p}\times 3\right)$ array containing the values of the three objectives; second, an $\left(N_{p}\times 22\right)$ array with the corresponding portfolio weights, which have been rounded to three decimal places. The computation of the ε-approximate Pareto front required a time of 34 h.

Next, in the a-posteriori approach we obtain the bounded ε-approximate Pareto front according to the following carbon risk and risk aversion related preferences:Concerning the carbon risk related preferences, the green investors are classified into three profiles according to expression (Equation 11) by using the 25th percentile for the Strong green investor $p_{g}^{s}=P_{25}$, the 55th percentile for the Moderate green investor $p_{g}^{m}=P_{55}$, and the 75th percentile for the Weak green investor $p_{g}^{w}=P_{75}$. Thus, the reference vector for the green investor is obtained by applying these percentiles to the column of Carbon risk at the $\left(N_{p}\times 3\right)$ array containing the ε-approximate Pareto front:
$$P_{g}=\left[P_{25},P_{55},P_{75}\right]=\left[2.803,3.001,3.135\right]$$Considering the investor’s loss aversion attitude, the investors are classified into three profiles according to expression (Equation 12) by establishing the 50th percentile for a Conservative investor $p_{r}^{c}=P_{50}$, the 75th percentile for a cautious investor $p_{r}^{k}=P_{75}$, and the 100th percentile for an Aggressive investor $p_{r}^{a}=P_{100}$. Thus, the reference vector regarding the risk aversion is obtained by applying these percentiles to the column of Risk at the $\left(N_{p}\times 3\right)$ array containing the ε-approximate Pareto front:
$$P_{r}=\left[P_{50},P_{75},P_{100}\right]=\left[9.575,10.095,11.633\right]$$

From Table 2, Table 3 and Table 4, a comparison of efficient portfolios for Weak, Moderate and Strong green investors is made in terms of different loss aversion attitude. To this end, for each profile, we display a numerical description of the portfolio composition ($F_{i}=100\;\cdot \;\omega_{i}$) and the objective function values attained by the portfolios. Figure 2, Figure 3 and Figure 4 show the 3D representation of the ε-approximate Pareto front, thus providing the non-dominated mean-variance-emission surface for the three types of Green investor profile and each level of loss aversion. Notice that, as the level of loss aversion attitude decreases, the Green Investor non-dominated surface (colored in blue) grows.

## 5. Discussion

The results are displayed in Table 2, Table 3 and Table 4, for each green investor profile. Each table contains three groups of rows corresponding to the risk aversion profile, and two groups of columns with the information about the funds’ portfolio composition ($F_{i}=100\;\cdot \;\omega_{i}$) and the objective function values attained by the portfolios [Risk, Ret., Carb.]. Given a green investor profile and a risk aversion profile, we consider four strategies as follows:In the first row, we highlight in bold optimal funds allocation for the nearest solution in the ε-approximate Pareto front to the ideal values of return, risk, and carbon risk, when the investor is seeking the optimization of the three objectives simultaneously [opt].In the second row, we display the optimal funds’ allocation when the investor is focused on achieving the minimum risk [min risk].In the third row, we report the optimal funds’ allocation when the investor tries to minimize the carbon risk exposure of the portfolio [min carb].In the fourth row, we display the optimal funds allocation when the investor seeks to maximize the return of the portfolio [max ret].

As a complement to the information supplied by above mentioned tables, we add Figure 2, Figure 3 and Figure 4. Thus, each figure is associate to a green investor profile and includes three representations of the ε-approximate Pareto front for the corresponding risk aversion profile: (a) conservative, (b) cautious, and (c) aggressive.

Let us see, for example, the case of a Weak green investor with a conservative attitude toward risk, which information is provided in Table 2 and Figure 2a. If the investor is seeking the optimization of the three objectives simultaneously, the optimal portfolio allocation is given by $F_{3}$, $F_{10}$, $F_{11}$, $F_{12}$, $F_{14}$, $F_{16}$, and $F_{21}$ [opt]. We can also view the 3D non-dominated surface, Figure 2a, in which the whole ε-approximate Pareto front is colored in grey, and the bounded investor’s region of interest is faded from blue to green. While the nearest solution to the ideal values for the three objectives lies at the center of the bounded region of interest [opt], the corner solutions indicate the objective values involving minimum risk [min risk], minimum carbon risk [min carb], and maximum return [max ret]. A red dot marks these optimal values. Note that the bounded region of interest increases as the investor’s risk aversion decreases (see Figure 2b,c). We can observe that the more aggressive attitude toward risk, the more returns and the lower carbon risk values are reached.

When analyzing the Moderate and Strong green investor results, displayed in Table 3–Figure 3 and Table 4–Figure 4, respectively, we would like to remark the same behavior as explained for the Weak green investor. Notice that, as the level of loss aversion attitude decreases, the bounded ε-approximate Pareto front (colored in blue–green) grows.

Comparing data through Table 2, Table 3 and Table 4, we stress at this point that the lower level of returns corresponds to a Strong green investor regardless of the investor’s risk aversion preferences, as shown in Figure 4. As expected, the minimum region of interest corresponds to a Strong green investor with a conservative attitude toward risk, as shown in Figure 4a. We also would like to remark that a Moderate green investor with an Aggressive attitude toward risk can achieve a high level of return (see Figure 3c). Thus, we conclude that green investors have a leeway to decrease the emission risk of the portfolio at no cost to risk and returns.

## 6. Conclusions

In this paper, we have proposed a new application of the ev-MOGA algorithm to handle a tri-criterion portfolio optimization problem in which the third criterion is the carbon risk score of the portfolio. Moreover, we have incorporated the investor’s preferences regarding the risk emissions and the loss aversion attitude into the solution process by defining different investor profiles. This allows us to propose a solution to the investor in terms of their carbon risk and risk aversion related preferences.

Given the urgency around climate change, investors are becoming increasingly aware of the need to make the transition to a low carbon economy and to address climate-change related risks. To the best of our knowledge, this is the first time that a MOEA based on the concept of ε-dominance called ev-MOGA has been used to assess and manage the impact of climate-related risks in the portfolio selection problem. The ev-MOGA allows us to derive a 3D-Pareto front in a well-distributed manner with limited memory resources. Next, we have introduced a-posteriori approach to include the investors’ preferences about green investments and risk aversion to understand the trade-off between the three objectives. So, by considering different investor profiles, we can provide a more approximated solution according to their preferences. In applying this approach to a sample of institutional SRI European open-end funds, we obtain a global ε-approximate Pareto front including a vast region of solutions. The a-posteriori analysis shows the bounded ε-approximate Pareto front for each investor profile and, we find that the more aggressive attitude towards risk, the more returns and the lower the carbon risk values, namely, aggressive investors looking for high returns are allowed to invest in funds with a lower level of carbon risk scores. Moreover, we can also conclude that green investors have a leeway to decrease the carbon risk of the portfolio at no cost to risk and returns.

Because of the possibility to obtain an efficient frontier in three dimensions while including the choices on risk and climate-change related risks, we believe this is a useful tool for investors, especially for those who are willing to rebalance their portfolios towards more climate-conscious firms.

## Figures and Tables

**Figure 1 ijerph-17-06324-f001:**
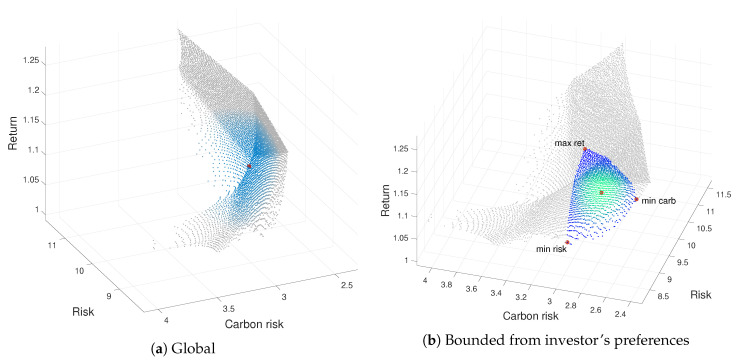
Approximated ε-Pareto front generated by ev-MOGA algorithm for the tri-criterion problem.

**Figure 2 ijerph-17-06324-f002:**
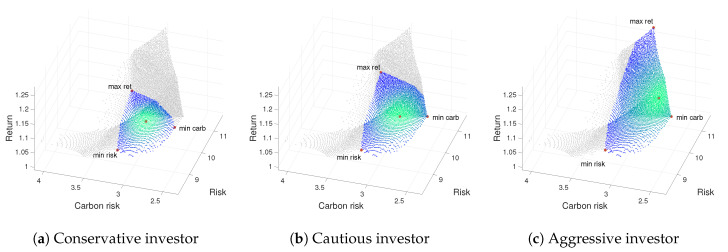
3D Pareto fronts for a Weak green investor profile ($p_{g}^{w}$): on the left, for the conservative profile ($p_{r}^{c}$); on the center for the cautious investor ($p_{r}^{k}$); and on the right the aggressive profile ($p_{r}^{a}$).

**Figure 3 ijerph-17-06324-f003:**
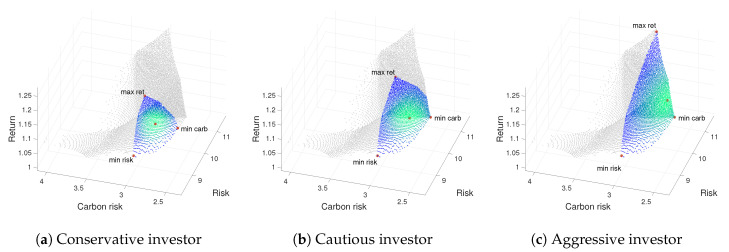
3D Pareto fronts for a Moderate green investor profile. ($p_{g}^{m}$): on the left, for the Conservative profile ($p_{r}^{c}$); on the center for the Cautious investor ($p_{r}^{k}$); and on the right for the Aggressive profile ($p_{r}^{a}$).

**Figure 4 ijerph-17-06324-f004:**
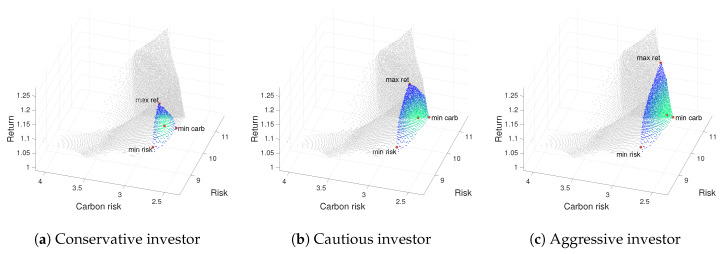
3D Pareto fronts for a Strong Green Investor profile ($p_{g}^{s}$): on the left, for the conservative profile ($p_{r}^{c}$); on the center for the cautious investor ($p_{r}^{k}$); and on the right for the aggressive profile ($p_{r}^{a}$).

**Table 1 ijerph-17-06324-t001:** Parameter setting of the ev-MOGA.

Parameter	Value
Size of the main population	$N_{\mathrm{indP}}=10^{4}$
Size of the auxiliary population	$N_{\mathrm{indGA}}=500$
Maximum algorithm iterations	$k_{\max}=10^{5}$
Probability of crossing/mutation	$P_{\;}c\;/\;m=0.20$
Number of boxes	$n_{\mathrm{box}}=300$
Maximum portfolio weights	$\omega_{\max}=0.20$

**Table 2 ijerph-17-06324-t002:** Weak green investor portfolio composition and objective value function.

Risk Profile	$F_{3}$	$F_{10}$	$F_{11}$	$F_{12}$	$F_{13}$	$F_{14}$	$F_{16}$	$F_{21}$	Risk	Ret.	Carb.	
Conservative	**20.0**	**20.0**	**20.0**	**12.3**	**0.0**	**6.5**	**18.6**	**2.6**	**9.106**	**1.142**	**2.871**	**opt**
	20.0	20.0	19.5	2.2	0.0	1.6	16.7	20.0	8.459	1.079	3.129	min risk
	20.0	20.0	16.0	0.0	0.0	19.6	20.0	4.4	9.572	1.099	2.581	min carb
	20.0	13.7	20.0	20.0	8.9	0.0	17.4	0.0	9.565	1.202	3.118	max ret
Cautious	**20.0**	**20.0**	**20.0**	**12.0**	**0.0**	**9.9**	**18.1**	**0.0**	**9.312**	**1.148**	**2.778**	**opt**
	20.0	20.0	19.5	2.2	0.0	1.6	16.7	20.0	8.459	1.079	3.129	min risk
	20.0	20.0	20.0	0.0	0.0	20.0	20.0	0.0	9.801	1.123	2.506	min carb
	20.0	6.1	20.0	20.0	12.9	1.4	19.6	0.0	10.059	1.228	3.134	max ret
Aggressive	**20.0**	**11.1**	**20.0**	**14.7**	**0.0**	**18.1**	**16.1**	**0.0**	**9.881**	**1.174**	**2.678**	**opt**
	20.0	20.0	19.5	2.2	0.0	1.6	16.7	20.0	8.459	1.079	3.129	min risk
	20.0	20.0	20.0	0.0	0.0	20.0	20.0	0.0	9.801	1.123	2.506	min carb
	20.0	0.0	20.0	20.0	20.0	20.0	0.0	0.0	11.633	1.271	3.014	max ret

**Table 3 ijerph-17-06324-t003:** Moderate green investor portfolio composition and objective value function.

Risk Profile	$F_{3}$	$F_{10}$	$F_{11}$	$F_{12}$	$F_{13}$	$F_{14}$	$F_{16}$	$F_{21}$	Risk	Ret.	Carb.	
Conservative	**20.0**	**20.0**	**20.0**	**8.5**	**0.0**	**9.0**	**19.3**	**3.2**	**9.191**	**1.133**	**2.805**	**opt**
	20.0	20.0	12.9	0.4	0.0	8.9	18.0	19.8	8.652	1.053	2.995	min risk
	20.0	20.0	16.0	0.0	0.0	19.6	20.0	4.4	9.572	1.099	2.581	min carb
	20.0	11.7	20.0	20.0	4.7	3.8	19.8	0.0	9.552	1.194	2.994	max ret
Cautious	**20.0**	**20.0**	**20.0**	**8.7**	**0.0**	**11.9**	**19.4**	**0.0**	**9.409**	**1.141**	**2.712**	**opt**
	20.0	20.0	12.9	0.4	0.0	8.9	18.0	19.8	8.652	1.053	2.995	min risk
	20.0	20.0	20.0	0.0	0.0	20.0	20.0	0.0	9.801	1.123	2.506	min carb
	20.0	4.6	20.0	20.0	8.6	6.8	20.0	0.0	10.081	1.219	2.992	max ret
Aggressive	**20.0**	**12.0**	**20.0**	**11.7**	**0.0**	**20.0**	**16.3**	**0.0**	**9.950**	**1.166**	**2.622**	**opt**
	20.0	20.0	12.9	0.4	0.0	8.9	18.0	19.8	8.652	1.053	2.995	min risk
	20.0	20.0	20.0	0.0	0.0	20.0	20.0	0.0	9.801	1.123	2.506	min carb
	19.9	0.0	20.0	20.0	19.0	19.7	1.4	0.0	11.529	1.267	3.001	max ret

**Table 4 ijerph-17-06324-t004:** Strong green investor portfolio composition and objective value function.

Risk Profile	$F_{3}$	$F_{10}$	$F_{11}$	$F_{12}$	$F_{13}$	$F_{14}$	$F_{16}$	$F_{21}$	Risk	Ret.	Carb.	
Conservative	**20.0**	**20.0**	**20.0**	**2.6**	**0.0**	**13.3**	**19.9**	**4.2**	**9.353**	**1.118**	**2.698**	**opt**
	20.0	19.9	10.9	0.6	0.0	15.5	19.5	13.6	9.054	1.062	2.795	min risk
	20.0	20.0	16.0	0.0	0.0	19.6	20.0	4.4	9.572	1.099	2.581	min carb
	20.0	13.5	20.0	18.0	0.0	12.7	15.8	0.0	9.571	1.175	2.794	max ret
Cautious	**20.0**	**20.0**	**20.0**	**4.6**	**0.0**	**15.9**	**19.5**	**0.0**	**9.588**	**1.133**	**2.613**	**opt**
	20.0	19.9	10.9	0.6	0.0	15.5	19.5	13.6	9.054	1.062	2.795	min risk
	20.0	20.0	20.0	0.0	0.0	20.0	20.0	0.0	9.801	1.123	2.506	min carb
	20.0	3.6	20.0	20.0	2.2	14.2	20.0	0.0	10.084	1.204	2.794	max ret
Aggressive	**20.0**	**20.0**	**20.0**	**4.6**	**0.0**	**19.6**	**15.8**	**0.0**	**9.719**	**1.135**	**2.571**	**opt**
	20.0	19.9	10.9	0.6	0.0	15.5	19.5	13.6	9.054	1.062	2.795	min risk
	20.0	20.0	20.0	0.0	0.0	20.0	20.0	0.0	9.801	1.123	2.506	min carb
	20.0	0.0	20.0	20.0	7.3	19.6	13.1	0.0	10.716	1.229	2.803	max ret

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
