# Peer review of "Tri-Criterion Model for Constructing Low-Carbon Mutual Fund Portfolios: A Preference-Based Multi-Objective Genetic Algorithm Approach"

_ijerph, 2020, doi:10.3390/ijerph17176324_

Round 1
Reviewer 1 Report
My comments:
1. This paper constructs “a preference-based multi-objective genetic algorithm approach” that contributes a new way in academic research and practice of constructing low-carbon mutual fund portfolios.
- It is rigorous and appropriate for author(s) using a preference-based multi-objective genetic algorithm approach—the formulas are well developed and organized.
- The methodology of this paper is rigorous and scientific.
- The analysis and statement of results is correct and sufficient.
- It is necessary to reduce the length of introduction, discussion, and conclusions.
- The “Conclusion” must be refined and supplemented for the contributions to academic research and theoretical implications of this study.
Reviewer 2 Report
I have reviewed the manuscript entitled "Tri-criterion model for constructing low-carbon mutual fund portfolios: a preference-based multi-objective genetic algorithm approach." The paper has merit because it is well written, it is adequately directed to the target audience, and the algorithm and the results are clearly explained (the plots are very well presented. Congrats!).
However, there are some drawbacks to address:
- This study is an extension of a previous publication by the same authors (see doi.org/10.1155/2019/6095712); this is not necessarily a bad situation, but the authors must clarify the specific contribution of this paper in comparison with the previous one. Some long explanations about evMOGA are provided; however, in this paper, they must be briefly written in a scientific style, and the authors should cite previous studies for further details. So, the level of similarity between both papers could be diminished. In its current state, readers hardly could identify significant differences.
- The authors claim:
"Our paper makes the following contributions to the literature. First, it gives a better understanding of recent multi-criteria decision-making methodologies (MCDM) to deal with tri-criterion portfolio selection problems by reviewing the literature on exact methods and multi-objective genetic algorithms techniques" (Lines 53-55).
Although the literature review is sufficient for the paper, it is inadequate to be a contribution alone. I think the authors can write another article (a survey or a comprehensive review) that does achieve this aim.
- Definition 4 is confusing; it seems to favor ε-dominated solutions, which is wrong, is there a mistake here? Could the authors provide a clearer definition?
- In Section 4, from where did the authors get the raw values of P_g and P_r? It is not specified before.
- In Section 4, there is no information on run times (how much time is required to get solutions?) and the experimental environment (what type of computer did the authors use? an average one?). From a practical point of view, this information is meaningful.
- When gender is unknown, the authors should use the singular 'they' instead of 'he/she' because decision-makers are often a group of people rather than only one person. Besides, even if the latter case occurs, gender must not be binarized (to favor social inclusion).
- In the title, I think the word "approach" is unnecessary.
I have also attached a PDF file with specific suggestions.

Reviewer 3 Report
This is a very interesting paper proposing a multi-criteria portfolio optimization approach, which takes into account the carbon score as one of the used criteria for optimization. In my opinion, the paper would still benefit from some amendments:
- The paper is generally written very well, however, it still entails minor typos and errors; Another round of proof-reading would be very helpful. A few instances are provided below, though please check the entire manuscript:
Line 29; require paraphrasing (as regard sustainability)
Line 34; the full wording of the acronym should be ‘Multicriteria decision making’, not ‘problem’.
- Heading number 2 is quite wordy; I’d suggest that you shorten the heading please.
- l believe you could extend the literature by discussing other available techniques for portfolio risk approximation e.g. copula.
- In the earlier sections, authors should highlight that the Sustainalytics carbon measure is the same as the Morningstar carbon score, which is mentioned under section 4 (I believe the former is now part of Morningstar). This inconsistency in wording may cause confusion for some readers.
- The methodology is very interesting and results are presented well. Nonetheless, the paper could benefit from further discussions on the results. I would suggest that the authors open up a new section titled ‘discussion’ to provide a critical discussion of their findings, and related implications. Also, I believe the second paragraph in the ‘conclusion section’ could be moved into the newly created discussions section.
- Most of references are dated and would require updating.
